# Dissecting Toxicity: The Venom Gland Transcriptome and the Venom Proteome of the Highly Venomous Scorpion *Centruroides limpidus* (Karsch, 1879)

**DOI:** 10.3390/toxins11050247

**Published:** 2019-04-30

**Authors:** Jimena I. Cid-Uribe, Erika P. Meneses, Cesar V. F. Batista, Ernesto Ortiz, Lourival D. Possani

**Affiliations:** 1Departamento de Medicina Molecular y Bioprocesos, Instituto de Biotecnología, Universidad Nacional Autónoma de México, Avenida Universidad 2001, Cuernavaca, Morelos 62210, Mexico; jcidu@ibt.unam.mx; 2Laboratorio Universitario de Proteómica, Instituto de Biotecnología, Universidad Nacional Autónoma de México, Avenida Universidad 2001, Cuernavaca, Morelos 62210, Mexico; pkas@ibt.unam.mx (E.P.M.); fbatista@ibt.unam.mx (C.V.F.B.)

**Keywords:** *Centruroides limpidus* Karch, proteome, scorpion, transcriptome, venom toxicity

## Abstract

Venom glands and soluble venom from the Mexican scorpion *Centruroides limpidus* (Karsch, 1879) were used for transcriptomic and proteomic analyses, respectively. An RNA-seq was performed by high-throughput sequencing with the Illumina platform. Approximately 80 million reads were obtained and assembled into 198,662 putative transcripts, of which 11,058 were annotated by similarity to sequences from available databases. A total of 192 venom-related sequences were identified, including Na^+^ and K^+^ channel-acting toxins, enzymes, host defense peptides, and other venom components. The most diverse transcripts were those potentially coding for ion channel-acting toxins, mainly those active on Na^+^ channels (NaScTx). Sequences corresponding to β- scorpion toxins active of K^+^ channels (KScTx) and λ-KScTx are here reported for the first time for a scorpion of the genus *Centruroides*. Mass fingerprint corroborated that NaScTx are the most abundant components in this venom. Liquid chromatography coupled to mass spectometry (LC-MS/MS) allowed the identification of 46 peptides matching sequences encoded in the transcriptome, confirming their expression in the venom. This study corroborates that, in the venom of toxic buthid scorpions, the more abundant and diverse components are ion channel-acting toxins, mainly NaScTx, while they lack the HDP diversity previously demonstrated for the non-buthid scorpions. The highly abundant and diverse antareases explain the pancreatitis observed after envenomation by this species.

## 1. Introduction

Scorpion venoms are known to contain hundreds of pharmacologically active components, affecting many other animals, which constitute their preys, competitors and/or predators [1]. They have a cosmopolitan distribution with 2415 distinct species reported up to December 2018 [2], which are classified into twenty different families. The scorpions belonging to the family Butidae produce the most active toxins that affect mammals, including humans [3,4]. Mexico harbors approximately 12% of the world’s diversity of scorpions, with 281 different species thus far described, of which 21 are dangerous to humans [5]. Their medical importance drove the initial research towards the isolation and biochemical characterization of the toxic compounds present in their venoms, and the identification of their physiological effects. The more recent development of high-throughput methods for massive sequencing and mass spectrometry has impacted the number of identified venom components [6,7,8,9,10,11,12,13].

In Mexico, the scorpions dangerous to humans belong to the genus *Centruroides*, comprising 42 different species [4,5]. The best studied species are: *Centruroides noxius*, *Centruroides suffusus*, *Centruroides tecomanus*, and *Centruroides limpidus*. *Centruroides limpidus* (Karsch, 1879) (Figure 1A) is widely distributed in densely populated areas of Central Mexico, including the States of Guerrero, Morelos, Mexico State, Michoacán, Queretaro, Hidalgo, and Puebla (Figure 1B) [14], and has a tendency to live side by side with humans [15]. It produces a potent venom which is highly toxic for mammals (the LD_50_ in mice is approximately 15 μg/20 g [16]) and poses a serious threat to human life. Even though it is practically impossible to single out the species involved in each envenomation case, *C. limpidus* and other closely related species of the *Centruroides* genus with which it shares its habitat, are responsible for over 120,000 reported accidents with humans in those states alone every year—a third of all scorpionism cases in the country [17]. Those high morbidity numbers alone justify the efforts made toward the characterization of the venom components responsible for human intoxication. The identification of the toxins present in this venom should impact, for example, the research aimed at the production of antivenoms [18]. Thus far, for *C. limpidus* alone, nine peptides have been biochemically characterized, and nine precursors for other putative toxins have been described [19,20,21,22,23]. It is clear, however, that the characterization of this venom is far from complete, as information on novel, potentially lethal toxins from this species continues to emerge [23]. A more comprehensive study was therefore required.

This communication reports the transcriptomic and proteomic analyses of the venom glands and the soluble venom of the scorpion *C. limpidus*, respectively. The transcriptomic results showed 192 transcripts encoding proteins/peptides with sequences identified as authentic venom components. Among these sequences, five main categories of precursors were identified: toxins, host defense peptides (HDPs), protease inhibitors, enzymes, and other components. The peptides encoded by the 46 transcripts identified in the transcriptome analysis were confirmed to be expressed in the venom by LC-MS/MS sequencing. They showed sequence similarity with previously reported scorpion venom components.

The relative abundance and diversity of ion channel-acting toxins, in particular those active on Na^+^ channels, is reported, confirming their responsibility in venom toxicity for mammals. The presence of several transcripts coding for antareases provide molecular support for the observed pancreatitis resulting from this species’ envenomation. Toxins from families never before identified in *Centruroides* venoms are also reported.

## 2. Results and Discussion

### 2.1. RNA Isolation, Sequencing, and Assembly

Two groups of scorpions separated by gender (5 males and 5 females) were used for total RNA isolation. A total of 9 µg (males) and 8 µg (females) of RNA was obtained and its quality was assessed with a Bioanalyzer 2100 (Agilent), as reported in other transcriptomic analyses [7,10,12,24,25]. The results indicated that the RNA samples were not degraded, even though a single RNA band corresponding to the mitochondrial 18S was observed, an effect previously reported for other organisms [7,10,12,24,25]. Paired-end cDNA libraries were prepared for males and females separately and sequenced using the Illumina platform (2 × 72 bp reads). The two genders were sequenced independently to fulfill the requirements of a related project that focuses on the differential expression of venom components in both genders [26], and which the results of will be published elsewhere. Two datasets with 38,364,311 and 41,366,601 reads were obtained for males and females, respectively, which were submitted to the European Nucleotide Archive (ENA), under project PRJEB31683. In order to have a global, species–specific transcriptomic analysis, comparable with other published scorpion transcriptomes, both datasets were merged and jointly analyzed henceforth. After de novo RNA-seq assembly with the Trinity software, 198,662 putative transcripts were obtained, with an N50 of 1611. The annotation was performed with the Trinotate software, resulting in 11,058 annotated transcripts that were identified by sequence similarity with sequences deposited in the Uniprot database. The divergence between the large number of assembled transcripts and the smaller subset of sequences with annotation, reflects the lack of information on many scorpion venom components, and reinforces the need for further biochemical and functional characterization of the scorpion venoms. Among the annotated transcripts, 366 sequences had similarity with arachnid sequences, 227 were specific for scorpions, and 192 sequences were from components related to venom, in particular. The 192 venom-related sequences were analyzed by BLAST to identify the closest related matches by sequence. The transcripts described in this study were labeled following the previously suggested standard for transcript nomenclature [10]. The species code for *C. limpidus* was set to “Cli”, the existing family and subtype codes were observed, and new ones were added for components not previously found in scorpion venoms. A complete listing of the 192 sequence IDs, organized by type/subtype, the ID of the closest matches (with E values), and their translated Open Reading Frame (ORF) is provided in Appendix A. For the reference sequences, the original names found in the databases were honored.

### 2.2. The Diversity of Transcripts Related to Venom Components in the Venom Gland of *C. limpidus*

The identification of transcripts putatively coding for venom peptides/proteins in *C. limpidus* was initially based on Pfam domains [27]. The search for matching domains was not limited to scorpions, but broadened to include all venomous animals [28]. Four major categories were identified by this method: toxins, host defense peptides, enzymes, and protease inhibitors. A fifth heterogeneous category denominated as “other components” was designated to include annotated sequences without defined domain and function, as well as sequences with well-conserved domains, but without a defined function in venoms. Figure 2 shows the distribution of the annotated venom-related sequences grouped into these five categories, as percentages of the total 192. For this figure, the number of individual sequences was considered, rather than their relative abundance in the transcriptome (and therefore the term “Diversity” was used in the graph). Each major category was further divided into subcategories, determined by the more specific, putative structural or functional classification of the peptides/proteins. The transcripts were also globally classified in accordance to Gene Ontology (GO) terms [29] (Appendix A).

#### 2.2.1. Ion Channel-Acting Toxins

Hundreds of toxic components have been described to be present in scorpion venoms [1]. Many of these are peptides known to affect ion channels from mammals, birds or arthropods (e.g., arachnids, insects or crustaceans), including sodium, potassium, calcium and chloride ion channels [30].

As shown in Figure 2, this category was the most diverse in terms of sequence in the transcriptome of *C. limpidus*, which did not come as a surprise, given the known high toxicity of this species to animals from different taxa. Eighty-five distinct sequences with conserved motifs or domains for ion channel-acting toxins were found, including 59 transcripts potentially coding for sodium channel-acting toxins (NaScTx) and 26 for putative potassium channel-acting toxins (KScTx). This is by far the largest number of toxin sequences reported for a scorpion of any species to date. Appendix A contains all the transcripts putatively coding for ion channel-acting toxins, together with the reference protein/transcript (the best match by sequence similarity), the source of the reference, plus its function if known.

##### Toxins Active on Voltage-Gated Sodium Channels (NaScTx)

Due to their central role in the intoxication process following envenomation, the scorpion toxins that affect voltage-gated sodium channels are the best characterized venom components, both biochemically and functionally. They are broadly subdivided into α-NaScTx and β-NaScTx, the former being known for slowing down the Na^+^ channel inactivation and binding to receptor site-3, while the latter shifts the channels’ opening kinetics to more negative potentials and bind to receptor site-4 [31]. Scorpion β-NaScTx are further classified as classical (active on mammalian Na^+^ channels), excitatory anti-insect, depressant anti-insect, and β-like (compete for binding sites on both insect and mammalian Na^+^ channels) toxins [32].

The genome sequences from a scorpion of the same genus, *Centruroides sculpturatus*, have been recently released as BioProject PRJNA422877 on NCBI. As expected, many of the transcripts recovered in this study had sequence similarity with sequences from *C. sculpturatus*. Sequences matching those of other scorpions were also found, not only of the *Centruroides* genus, but also from old-world scorpions of the genera *Isometrus*, *Lychas* and *Parabuthus*. In the particular case of the putative NaScTx, 45 transcripts were found to be similar to sequences from *C. sculpturatus*, seven to previously reported sequences from *C. limpidus*, three to those from *C. noxius*, and one to a similar sequence from each of the following species, *C. exilicauda*, *Centruroides vittatus*, *Parabuthus transvaalicus*, and *Lychas mucronatus* (Appendix A).

The NaScTx are usually more abundant in scorpions of the family Buthidae, which produce highly neurotoxic venoms. Three previous high-throughput sequencing-derived transcriptomic analyses with scorpions belonging to this family (*Tityus bahiensis*, *Centruroides hentzi*, and *C. noxius* [6,8,33]), reported 27 to 38 transcripts potentially coding for NaScTx. This study found 59 transcripts of this kind, which represents the largest diversity so far described for scorpions of this, or any other taxonomic family. Of them, 16 transcripts putatively code for α-NaScTx and 43 for β-NaScTx (Appendix A).

Figure 3A shows, as an example, a multiple-sequence alignment of CliNaTAlp03 and CliNaTAlp08, both annotated as possible α-NaScTx, with their best matches in terms of sequence similarity. Only the sequence region corresponding to the predicted mature toxin was used for the alignment, since the signal and propeptide regions were missing in the sequences of the peptides obtained directly from the venom. The peptides encoded by CliNaTAlp03 and CliNaTAlp08 were confirmed to be present in the venom by the proteomic analysis, and were the α-NaScTx with the highest identification scores by that analysis. CliNaTAlp03 matched with “precursor alpha-like toxin CsEv5” (XP_023210703) and the peptide itself (P58779) [34]. CliNaTAlp08 matched with “precursor alpha-toxin CsE5” (XP_023242920) and the peptide itself (P46066) [35]. Here, and in all further alignments, the newly reported *C. limpidus* sequences are shown on top of the alignment for clarity.

Three transcripts, CliNaTxBet31, CliNaTxBet32, and CliNaTxBet33 are shown in Figure 3B, aligned to the best matching reference sequences (mature peptides only). CliNaTxBet31 codes for Cll2b (P59899), a peptide found in the *C. limpidus* venom [21], which was confirmed by the proteomic analysis in this study (see below). Cll2b is toxic to mice and active on sodium and calcium channels in cultured chick dorsal root ganglion cells [21]. CliNaTxBet32 potentially codes for a peptide similar to Cll2b, but with six amino acid changes. Cll2b is very close in sequence to Cll2 (P59898), one of the major toxins of the *C. limpidus* venom [18,23], having only two differences at the amino acid level. Cll2 is a highly neurotoxic peptide for mammals [19]. CliNaTxBet33 matches the precursor of Cll5b (Q7Z1K7) and was here confirmed to be present in the venom by the proteomic analysis. A very similar sequence, Cll5c (Q7YT61), which differs from Cll5b by just one residue, is also included in the alignment. The remaining 40 transcripts found in this study are described in Appendix A. These include four sequences with similarity to other previously reported sequences from *C. limpidus*.

##### Toxins Active on Potassium Channels (KScTx)

Potassium channels are the most widely distributed type of ion channels, found in basically all living organisms, where they play fundamental roles in the physiology of the cells [36,37,38]. Scorpion venoms contain toxins that affect these channels (KScTx), as demonstrated with insect and mammalian channels [37]. The KScTx have been classified into seven families (α, β, γ, δ, ε, κ, and λ) depending on their amino acid sequences, length, and 3D-structure [39,40,41]. They have been shown to be active on voltage-gated potassium channels (such as Kv1, Kv3, Kv4, Kv7, Kv11) and calcium-activated potassium channels (KCa1.1 or BK) [37].

This study reports 26 transcripts with sequence similarity to members of five families of KScTx: 15 sequences potentially coding for α-KScTx, 2 for β-KScTx, 3 for γ-KScTx, 3 for δ-KScTx, and 2 for λ-KScTx (Appendix A).

Figure 4A shows two examples of the peptide sequences derived from the transcriptomic analysis in this study, which potentially correspond to α-KScTx. CliKTxAlp15 codes for a peptide confirmed to be expressed in the venom of *C. limpidus* by the proteomic analysis here reported. It is similar to alpha-KTx4.5 from *Tityus costatus* (Q5G8B6), a toxin found to inhibit with low potency the Kv1.1, Kv1.2, Kv1.3, and Kv11.1 (ERG1) channels [42]. CliKTxAlp07 potentially codes for a peptide similar to Noxiustoxin-2 (Q9TXD1), having only one difference at the amino acid level. Noxiustoxin-2 has a paralyzing effect on crickets but is not toxic to mice or crustaceans [43]. Two other similar sequences were included in the alignment, A0A218QXG2, a nucleotide sequence reported from *Tityus serrulatus* (only the predicted, translated mature region was included), and Noxiustoxin (P08815), a toxin from *C. noxius* that blocks several Kv and KCa channels [44]. The remaining 14 transcripts found in this study are summarized in Appendix A, including CliKTxAlp10, which is similar to a previously reported toxin from this scorpion, CllTx1 (P45629).

One very relevant finding of this work is the description, for the first time, of potential β-KScTx in a scorpion from the *Centruroides* genus. Toxins of this family have been found in the venoms of buthid and non-buthid scorpions from the genera *Androctonus*, *Euscorpiops*, *Heterometrus*, *Hoffmannihadrurus*, *Liocheles*, *Mesobuthus*, *Pandinus*, and *Tityus*, but never before in *Centruroides*. Toxins of the β-KScTx family are active on Kv1.1, Kv1.3 or Kv4.2 mammalian channels [45]. Two transcripts were found in the venom gland of *C. limpidus* coding for peptides that were confirmed to be expressed in the venom by the proteomic analysis. CliKTxBet01 displayed sequence similarity with a DNA sequence from *C. sculpturatus* (XP_023220228) and with toxin TdiKIK (Q0GY43) from *Tityus discrepans*. CliKTxBet02 was similar to another DNA sequence from *C. sculpturatus* (XP_023220230) and to the “Scorpine-like peptide Tco 41.46-2” from *T. costatus*. The putative β-KScTx here described are shown aligned to the reference sequences (only the predicted mature sequences) in Figure 4B.

The γ-KScTx are short-chain peptides of 36–47 amino acid residues. Structurally, they have a cysteine-stabilized αβ (CSα/β) motif, with three or four disulfide bridges. The γ-KScTx are capable of blocking the ERG K^+^ channels. Three new sequences, potentially coding for γ-KScTx are reported here (Figure 4). CliKTxGam01 codes for a peptide identical to CllErg1 (Q86QV0) isolated from the venom of *C. limpidus*, though never tested on ion channels. It is also closely related to CeErgTx5, from *Centruroides elegans*, differing in just one amino acid. CeErgTx5 is active on mammalian ERG K^+^ channels [46]. CliKTxGam02 is similar to CnErg1 (Q86QT3) identified in the venom of *C. noxius*, a peptide with activity on mammalian ERG K^+^ channels [47]. CliKTxGam03 is closest in sequence to “potassium channel toxin gamma-KTx 1.1-like” (XP_023241648), derived from the genome sequences of *C. sculpturatus* (only the predicted mature sequence was considered in the alignment in Figure 4C). The three putative sequences, here reported, contained the functionally relevant K13, involved in the interaction with the outer vestibule of the channel [48]. Two other residues, demonstrated to be relevant for the interaction of γ-KScTx with the channels, are Q18 and M35 [49]. Only CliKTxGam01 has those residues conserved (Figure 4C), which might have implications in the functionality of the peptides derived from CliKTxGam02 and CliKTxGam03 if they happen to be indeed expressed in the venom.

The δ-KScTx are peptides with 59–70 amino acid residues. They are structurally characterized by the presence of a CSα/β motif of the Kunitz type, stabilized by three disulfide bonds [50]. They display a dual activity: as serine protease inhibitors and as blockers of the Kv channels, mainly the Kv1.3, though other channels can also be weekly inhibited. Three different transcripts are here described, which are shown aligned to reference sequences in Figure 4D. CliKTxDel01 is similar to “Kunitz-type serine protease inhibitor BmKTT-2” (P0DJ50), a peptide from *M. martensii* which completely inhibits trypsin and blocks the murine Kv1.3. CliKTxDel02 shares sequence similarity with a genome sequence from *C. sculpturatus* labeled as “isoinhibitor K-like” (XP_023217495). CliKTxDel03 was found to be similar to a genome sequence from the spider *Paraestatoda tepidariorum* annotated as “hemolymph trypsin inhibitor B-like isoform X2” (XP_015905918). The genome-derived sequences used in the alignment in Figure 4 are limited to the predicted mature sequences.

It is very relevant that no transcripts coding for calcium channel-specific toxins of the calcin family (active on the ryanodine receptors of mammalian cardiac or skeletal muscle cells) were found in the analysis, while two transcripts are here reported for the phylogenetically related, insect-specific λ-KScTx family. The λ-KScTx are short (approximately 40 amino acids) peptides which adopt an inhibitor cystine knot (ICK) fold. Both CliKTxLam01 and CliKTxLam02 code for the first λ-KScTx ever described for the genus *Centruroides*. Their finding in a buthid scorpion confirms the proposition that calcins and λ-KScTx are specific for non-buthid families and the buthid family, respectively, being mutually exclusive in those venoms [51]. In order to confirm that CliKTxLam01 and CliKTxLam02 are indeed λ-KScTx and not the structurally related calcins, these two sequences were incorporated into a phylogenetic analysis with those in Reference [51] by Carlos Santibáñez-López (see Acknowledgements) and shown to group with other buthid λ-KScTx and not the non-buthid calcins (Appendix A). CliKTxLam01 has sequence similarity with “phi-buthitoxin-Hj1a” (F1CIZ6), derived from a transcript from *Hottentotta judaicus*. CliKTxLam02 matched by sequence similarity the “potassium channel blocker pMeKTx30-1” (A0A088DAF5), derived from a transcript from *Mesobuthus eupeus*. These sequences are shown aligned in Figure 4E. Only two tested K^+^ channel blockers belonging to the λ-KScTx family have been reported thus far: ImKTx1 from *Isometrus maculatus* (P0DJL0) [52] and Neurotoxin lambda-MeuTx from *M. eupeus* (P86399) [53]. They are also included in the alignment for reference.

#### 2.2.2. Host Defense Peptides (HDPs)

Arachnid venoms are rich sources of host defense peptides (HDPs) [54]. They are characterized by having a broad spectrum of biological activities, including antimicrobial [55,56,57], insecticidal [58], bradykinin-potentiating [59], antitumoral [60,61], and hemolytic [62], among others. Host defense peptides have been demonstrated to be abundant and highly diverse in the venoms of scorpions belonging to non-Buthidae families [7,24,25], though a previous study with high-throughput sequencing techniques also identified HDPs in the buthid *C. hentzi* [8]. Host defense peptides are divided into two categories: non-disulfide-bridged-peptides (NDBPs) and cysteine-stabilized β-sheet-rich peptides, which include the defensins and scorpines [54]. Though just one HDP (a defensin whose expression is induced in the hemolymph in response to septic injury) had been reported from *C. limpidus* [63], a mass fingerprint of the venom revealed components with lower molecular weights than KScTx, indicating the potential presence of HDPs [26]. This analysis confirms that conclusion. Ten transcripts coding for HDPs were found: six defensins, one NDBP-2, two NDBP-4, and one anionic peptide. Nevertheless, compared to the above referenced studies by massive RNA sequencing of non-buthid scorpions, the diversity here described for the HDPs was significantly lower, confirming the previous empiric observation that, contrary to the ion channel-acting toxins, HDP are more diverse in non-buthid scorpions than in buthids.

CliHDPDef01 is actually the transcript coding for Cll-dlp (Q6GU94), the previously reported hemolymph defensin. It is notable that the same defense peptide is expressed in both tissues. The remaining five defensin transcripts, CliHDPDef02–CliHDPDef06, have similarity to Defensin-1 (A0A0K0LBV1), a transcript from *Androctonus bicolor* [64]. Both CliHDPDef01 and CliHDPDef02 are shown in Figure 5A, aligned to the reference sequences. Since the complete precursors for all the sequences are available, they were used in the alignment and included in the calculated percentage of identity.

Transcripts with sequence similarity to members of families NDBP-2 and NDBP-4 were also found. CliHDPND201 had similarity to “venom toxin meuTx20” (A0A146CJE0) deduced from a transcript from *M. eupeus*, and with BmKbpp (Q9Y0X4), a peptide from *M. martensii* with antimicrobial and bradykinin-potentiating activities against bacteria and fungi [65]. The precursor sequence of these three molecules are shown aligned in Figure 5B. CliHDPND401 has similarity to ToAP2 (A0A1D3IXJ5), a transcript isolated from the venom of *Tityus obscurus*. Synthetic ToAP2 displayed antimicrobial activity, both in vitro and in vivo [66]. CliHDPND402 is similar to peptide TsAP2 (S6D3A7), an antibacterial, antifungal, anticancer, and hemolytic peptide isolated from *T. serrulatus* [61]. The precursor sequences of these NDBP-4 are shown aligned in Figure 5C.

A transcript for a putative anionic HDP without disulfide bridges is also reported. CliHDPAni01 shows similarities to the genome-derived sequence XP_023227050 from *C. sculpturatus*, and with TanP (GeneBank ID of the transcript: JK483720), a peptide isolated from the venom of *Tityus stigmurus* [67]. The predicted precursors translated from these sequences, are shown aligned in Figure 5D.

#### 2.2.3. Enzymes

Enzymes are essential components of many animal venoms. Thought more abundant in snakes [68,69], enzymes have been discovered in venoms from many taxa, including ants [70], jellyfish [71], wasps [72], spiders [73], and scorpions [74]. Scorpions, in particular, were shown to contain proteases [75], phospholipases [76], and hyaluronidases [77] in their venoms by classical studies. More recently, high-throughput transcriptomic analyses with venom glands have reported the existence of transcripts putatively coding for other enzymes never before reported in scorpion venoms, e.g., 5′nucleotidases in non-buthid scorpions [10,12,24].

The venom gland of *C. limpidus* was found in this study to be rich in terms of transcripts putatively coding for enzymes. Forty-nine different sequences potentially coding for enzymes are here reported. Thirty-eight corresponded to proteases, seven to phospholipases, three to 5’nucleotidases, and one to a putative hyaluronidase. All of them were annotated by sequence similarity to genome-derived sequences from *C. sculpturatus*. They are all reported in detail in Appendix A, together with the reference genomic sequences.

The large number of protease-encoding sequences in this scorpion is remarkable. It is particularly rich in transcripts for metalloproteases, with 24 out of the 38 transcripts potentially coding for proteases, being for metalloproteases (the remaining 14 are for serine proteases). It is also noticeable that the most diverse metalloprotease transcripts are those coding for antarease-type Zn-metalloproteases, 14 out of 24. Antarease was first described in the venom of *T. serrulatus* [78]. The envenomation by *Tityus* species can lead to the development of acute pancreatitis [79,80,81,82,83]. Venoms from *Tityus* scorpions are potent secretagogues that can elicit the release of secretory proteins from the pancreas [80,84,85,86]. Antarease was shown to specifically cleave the soluble N-ethylmaleimide-sensitive factor attachment protein receptors (SNAREs) involved in pancreatic secretion, disrupting the normal vesicular traffic [78]. Antarease could, therefore, be responsible for the acute pancreatitis induced by the *T. serrulatus* venom. Antarease-like enzymes were shown thereafter to be ubiquitous in the venoms of different scorpion genera [74]. The venom from *C. limpidus* was also shown to elicit manifestations associated with pancreatitis and to act as a secretagogue for amylase in the mouse pancreas [87]. The abundance of transcripts coding for antareases in *C. limpidus* could be an indicative of the prominent role of these enzymes in the manifestation of pancreatitis after scorpion envenomation by this species. The presence of antareases in the venom was here confirmed by the proteomic analysis. The proteins encoded by CliEnzMtp15, CliEnzMtp19, CliEnzMtp20, and CliEnzMtp21 were found by LC-MS/MS.

The other found transcripts, potentially coding for disintegrins, reprolysins, an astacin, and an angiotensin-converting enzyme complete the picture of the metalloproteases putatively expressed in this venom (Appendix A). The astacin (CliEnzMtp23) and the angiotensin-converting enzyme (CliEnzMtp24) were also confirmed to be present in the venom by LC-MS/MS.

It is intriguing that, even though transcripts potentially coding for phospholipases A2 and D2 were found in the transcriptomic analysis (Appendix A), none were confirmed by the proteomic analysis. The inability to detect them in the proteome seems to correlate with the absence of phospholipase activity in the *C. limpidus* crude venom [26] as assayed by the egg-yolk method of Haberman and Hard, which detects phospholipases A, B, and C [88]. Nevertheless, it should be kept in mind that this transcriptomic analysis is descriptive and not quantitative. Therefore, no inferences can be derived from the transcript detection about the transcript and protein levels in the gland and the venom, respectively.

An interesting family of enzymes, just recently identified in scorpion venoms are the 5´nucleotidases. They were initially found in snake venoms where they seem to be ubiquitous [89]. They are known to endogenously liberate purines, which potentiate venom-induced hypotension and paralysis via purine receptors [89]. In addition, they can synergistically interact with other venom proteins, contributing to the overall effects of venoms [89]. The presence of 5´nucleotidases in non-buthid scorpion venoms was demonstrated for *Paravaejovis schwenkmeyeri*, *Thorellius atrox*, and *Megacormus gertschi* [10,12,24]. This study confirms that this enzyme could also be present in a buthid scorpion venom—at least its mRNA is transcribed in the gland—suggesting that these enzymes could also be ubiquitous in scorpion venoms. Three transcripts for *C. limpidus* 5´nucleotidases are reported in Appendix A.

It was previously shown that the *C. limpidus* crude venom contains active hyaluronidases [26]. A single transcript encoding for a hyaluronidase, CliEnzHya01 (Appendix A), was found in the present study. Noticeably, its presence in the venom was confirmed by the proteomic analysis. By degrading hyaluronic acid from the extracellular matrix [90], this enzyme could facilitate venom spreading [91].

#### 2.2.4. Other Venom Components

Besides the ones described above, other components frequently found in scorpion venoms were also detected in this analysis (Appendix A).

Protease inhibitors, which, in principle, protect the venom components from autogenous degradation by venom proteases [69], though might also inhibit enzymes in the targets’ tissues with neurological consequences [92], were very diverse in terms of transcripts in *C. limpidus*. Nineteen precursors encoding potential protease inhibitors were found, 13 of the Ascaris-type [93], two of the Kunitz-type [94], and four serpins [95].

Ascaris-type protease inhibitors are serine protease inhibitors, which have a Trypsin-Inhibitor-Like (TIL) domain. They are long-chain peptides with approximately 60 amino acid residues, stabilized by five disulfide bonds [96]. Recombinant SjAPI was the first functionally characterized Ascaris-type protease inhibitor from animal venoms [93]. It inhibits chymotrypsin and elastase while being inactive on trypsin. Three transcripts described here (CliPInTIL01–CliPInTIL03) have sequence similarity with classical Ascaris-type protease inhibitors from scorpion venoms or transcriptomes. Ten others, (CliPInTIL04–CliPInTIL13), with conserved TIL domain, but which are larger (ca 14 kDa instead of the classical ca 6 kDa) and have more disulfide bridges than the classical inhibitors (up to five extra disulfides), are also reported. A fourteenth sequence with a TIL domain (annotated here as CliPInTIL14) was detected in the proteomic analysis but had no matching transcript. It was annotated within this group due to the presence of the TIL domain, but it is a much larger protein (ca 210 kDa).

Kunitz-type protease inhibitors have been described in scorpion venoms [94]. The transcripts here reported (CliPInKun01 and CliPInKun02) were similar to genome sequences from *C. sculpturatus*. Members of the superfamily of serine proteinase inhibitors (serpins) from venomous animals have been poorly studied [95,97]. Four transcripts encoding serpins, CliPInSrp01–CliPInSrp04 are here reported, which also have sequence similarities with genome sequences from *C. sculpturatus*.

The CAP superfamily proteins are well distributed in all organisms, where they display a variety of functions [98]. This superfamily includes three major groups of proteins: cysteine-rich secretory proteins (CRISP), antigen or allergen proteins from arthropod venoms, and pathogenesis-related proteins from plants (PR) [99]. The CRISP proteins and antigen/allergen proteins have been described in venoms [100]. There are reports showing that CRISP proteins can interact with ion channels [101,102]. Six precursors coding for CAP-like proteins are here reported, CliOthCAP01–CliOthCAP06 (Appendix A). They all code for putative proteins with a Pfam domain corresponding to the CAP superfamily and the transcripts are similar to genome sequences from *C. sculpturatus*. Both CliOthCAP02 and CliOthCAP05 were confirmed to be expressed in the venom by the proteomic analysis.

Transcripts for Insuline Growth Factor Binding Proteins (IGFBPs) have been found in scorpion venom glands [7,8,12,33,64,103,104], though the function of IGFBPs in venoms has not been established. Ten putative IGFBP transcripts are reported for *C. limpidus* (Appendix A). All of them matched genomic sequences from *C. sculpturatus*, except for CliOthIGF10, which was similar to a peptide annotated as “Venom toxin” (A0A1L4BJ69) from *Hemiscorpius lepturus*.

The first described La1 peptide was found in the venom of *Liocheles australiase* [105]. Peptides with similar amino acid sequences were thereafter annotated as “La1-like” peptides. They are usually long-chain peptides containing 73–100 amino acids, stabilized by four disulfide linkages, with a conserved single domain von Willebrand factor-type C (SVWC) structural motive. Transcripts for La1-like peptides have been identified in buthid [8] and non-buthid scorpions [7,10,12,24,104]. Six transcripts are here reported with those features, assumed to code for La1-like peptides (Appendix A). Of them, only CliOthLa106 was confirmed to be expressed as protein in the venom. Noticeably, CliOthLa106 is the only transcript among the six, which codes for a long-chain-type La1-like [24] (Appendix A).

Other orphan transcripts with ORF coding for putative peptides with no conserved structural domains, nor a known function, were also found. Since similar sequences had been described in other scorpion transcriptomic analyses or were directly identified in the *C. limpidus* venom, they were grouped as “undefined peptides” and annotated as CliOthUnd01–CliOthUnd05. Not much information can be provided on them, except for the reference sequences they are similar with (Appendix A).

### 2.3. Proteomic Exploration of the Venom Components of *C. limpidus*

#### 2.3.1. Mass Fingerprint of the Soluble Venom

The soluble fraction of the whole venom from sixty scorpions of mixed gender was used for the proteomic analysis. An aliquot of the soluble venom was applied to an HPLC coupled to a mass spectrometer, as previously reported [26]. The mass fingerprint detected 395 individual masses, ranging from 800 to 19,000 Da (Appendix A and Figure 6). The mass range with the largest number of individual masses detected was the one which spans 7001 to 8000 Da, which is within the expected range for the Na^+^ channel-acting scorpion toxins. This mass group was followed, in the number of independent masses, by the 4001 to 5000 Da range, which are the characteristic masses of K^+^ channel-acting scorpion toxins. This mass distribution corroborates the findings by the transcriptomic analysis pointing to the ion channel-acting toxins as the most diverse components in the venom of the highly toxic *C. limpidus*. It is relevant to note that, although transcripts potentially coding for enzymes were highly diverse in accordance to the transcriptomic analysis, the used setup for the mass fingerprint cannot precisely detect the mass of high molecular weight components (approximately above 10,000 Da) [24], so all those components are shown grouped in Figure 6.

#### 2.3.2. Identification of Peptides by LC-MS/MS

The peptides identified by LC-MS/MS, which also matched the sequences discovered by the transcriptomic analysis, were mentioned above in the description of the corresponding transcripts. This section summarizes those findings (Appendix A).

From a total of 52 identified sequences, 46 corresponded to molecular entities annotated as venom-related peptides/proteins. The remaining sequences corresponded to enzymes or proteins related to cellular processes, or without an identifiable conserved domain or structural motif. In correlation with the results of the transcriptomic and mass fingerprint analyses, the largest class of MS/MS-recovered sequences corresponded to toxins that affect ion channels, of which 26 were Na^+^ channel-acting toxins (7 α-type and 19 β-type, by sequence similarity) and three K^+^ channel-acting toxins. Tryptic peptides from eight enzymes were also recovered. Other venom components were also detected, but to a lesser extent. This includes two HDPs, one Ascaris-type protease inhibitor (though this is a protein much larger than the typical sequences of this kind, as discussed in the transcriptomic section), two proteins of the CAP superfamily, and one La1-like peptide of the long-chain type.

### 2.4. The Venom of the Highly Toxic *C. limpidus* versus the Venoms of Non-buthid Scorpions

Two other transcriptomic/proteomic analyses from scorpions of the genus *Centruroides* have already been published [6,8]. It would be interesting to compare these venoms in terms of their composition to try to define a possible pattern that would differentiate toxic from non-toxic species within this genus (*C. limpidus* and *C. noxius* are highly toxic to mammals while *C. hentzi* is not). However, even minor differences in sample preparation, cDNA library construction, cDNA sequencing protocol, MS/MS protocol or the bioinformatics analysis could lead to inaccurate conclusions. Nevertheless, the results here reported are comparable with those generated for other non-buthid scorpions under the exact same experimental protocol. That is the case for the analyses performed with *Serradigitus gertschi*, *Superstitionia donensis*, *T. atrox*, *P. schwenkmeyeri*, and *M. gertschi* [7,10,12,24,25]. As Figure 7A illustrates, the venom of *C. limpidus* is characterized by the highest diversity of toxins, in general. This is remarkable, considering also that no Ca^2+^ channel-acting toxins were recovered for the buthid species, so only two super-families, the NaScTx and the KScTx, are present in the venom. The fraction of recovered transcripts related to Na^+^ channel-acting toxins equals or surpasses the fraction of all toxins in the non-buthids, taken together. It is also relevant that the fraction of NaScTx in *C. limpidus* is larger than the fraction of KScTx, while for the non-buthids it is the other way around. It is well established that the toxicity of the scorpion venoms is primordially related to the neurotoxic action of the NaScTx [16,18]. It is well known that the neutralization of the main NaScTx in scorpion venoms results in the neutralization of the whole venom toxicity [106,107]. It is remarkable that, in the highly toxic *C. noxius*, the NaScTx are also more abundant and diverse than the KScTx [6,108], while in the non-lethal *C. hentzi*, the transcripts for NaScTx and KScTx are more or less equally diverse [8] (although, in absolute numbers, not comparable with the abundances here reported, as indicated above).

Figure 7B confirms that, on the contrary, the venoms from the non-toxic, non-buthid scorpions seem to be more diverse in terms of HDPs, with respect to the buthids, as proposed above in Section 2.2.2. Additionally, no direct link seems to exist between enzyme diversity and venom toxicity, according to the results charted in Figure 7C.

## 3. Conclusions

A total of 192 transcripts were identified in the present transcriptomic analysis. These sequences are assumed to code for Na^+^ and K^+^ channel-acting toxins, enzymes, HDPs, protease inhibitors, CAP-super-sfamily proteins, IGFBP, La-1-like peptides, and other orphan venom components. Mass fingerprint of the venom resulted in the detection of 395 individual components, the most abundant of which were peptides with molecular weights in the range of 7000 to 8000 Da, which are known to correspond to Na^+^ channel-acting toxins. The LC-MS/MS of the tryptically-digested venom confirmed that at least 46 of the venom-related, transcript-encoded proteins are indeed expressed in the venom.

The molecular dissection of the venom components from the highly toxic buthid scorpion *C. limpidus* revealed that the most abundant (from the mass fingerprint) and most diverse (from the transcriptomic and MS/MS analyses) venom components are the neurotoxic NaScTx. The fraction of toxins (most notably, the NaScTx) is significantly higher in *C. limpidus* than in other non-toxic, non-buthid scorpions from different genera. These findings correlate with classical biochemical and physiological observations on the relevance of the neurotoxic NaScTx in the toxicity of the scorpion venoms for mammals. It also reveals that the efforts directed at generating neutralizing antivenoms from toxin-specific human antibodies or their fragments might require a larger number of antibodies in the cocktail, depending on their effective cross-reactivity.

Molecular support for the observed pancreatitis after envenomation by this species is also provided. The relative abundance and diversity of the antarease-like Zn-metalloproteases seem to confirm their relevant role in this pathology.

Two families of toxins are described here for the first time in a scorpion of the genus *Centruroides*. Transcripts coding for β-KScTx were found, and the presence of the expressed toxins in the venom was confirmed by the proteomic analysis. Transcripts putatively coding for members of the λ-KScTx family are here also reported, though their presence in the venom remains to be demonstrated.

## 4. Materials and Methods

### 4.1. Biological Material

Adult scorpions of the species *C. limpidus* were collected in Morelos and Guerrero States, with official permit from the Secretaría de Medio Ambiente, Recursos Naturales y Pesca (SEMARNAT, numbers SGPA/DGVS/07805/16 and 004474/18). The collected specimens were maintained in plastic boxes with water ad libitum. Sixty adult scorpions were milked by electrical stimulation for the proteomic analyses. The venom was immediately suspended in deionized water and centrifuged at 15,000 *g* for 15 min. The protein content of the soluble venom was estimated with a Nanodrop 1000 (Thermo Fisher Scientific, Waltham, MA, USA) based on absorbance at λ = 280 nm, assuming that one absorbance unit equaled 1 mg/mL of protein. The soluble fraction of the whole venom was lyophilized and kept at −20 °C until used.

Seven days after milking, 5 male and 5 female scorpions were processed for telson dissection to extract the total RNA. These specimens were thereafter euthanized and preserved in ethanol as vouchers. The remaining scorpions were kept alive and released thereafter in the same locations where they were collected.

### 4.2. RNA Isolation, Sequencing, and Assembly

Total RNA was isolated as previously described [7,10,12]. The SV Total RNA Isolation System Kit (Promega, Madison, WI, USA) was used for this purpose. The telsons from 5 males and 5 females were dissected separately, under RNAse-free conditions, in microcentrifuge tubes containing the RNA lysis buffer. The optional 70 °C heating step of the protocol was followed before column purification. Total RNA was quantitated with a NanoDrop 1000 (Thermo Fisher Scientific, Waltham, MA, USA) and its quality was assessed with a 2100 Bioanalyzer (Agilent Technologies, Santa Clara, CA, USA).

Two cDNA libraries were prepared from 1 µg of total RNA for each gender, using the TruSeq Stranded mRNA Sample Preparation Kit (Illumina, San Diego, CA, USA), according to the manufacturer’s directions. DNA sequencing was performed at the Massive DNA Sequencing Facility of the Instituto de Biotecnología (Cuernavaca, Mexico) with a Genome Analyzer IIx (Illumina, San Diego, CA, USA) using a 72-bp paired-end sequencing scheme over cDNA fragments ranging 200–400 bp in size. The quality of the reads was verified with the FastQC program [109] after clipping off the adaptors.

The reads resulting from the sequencing of both male and female cDNA libraries were joined together for a de novo assembly into contigs using the Trinity software version 2.0.3 [110], employing the standard protocol. Basic statistics, such as the number of transcripts and contigs, were determined with the TrinityStats.pl script.

### 4.3. Bioinformatics

The assembled contigs were annotated as previously described [24], using the Uniprot/Uniref90 protein database for BLASTx and BLASTp. The prediction of ORFs were done with TransDecoderLongORfs. Putative signal peptides and propeptides were predicted with the ProP 1.0 server [111] and SpiderP from Arachnoserver [112]. Multiple sequence alignments were performed using MAFFT 7.0 online [113]. Alignments were edited in Bioedit [114] and in Adobe Illustrator CS6. All figures were generated with Rstudio [115]. The GO terms were quantified in WEGO [116].

### 4.4. Molecular Mass Fingerprint by LC-MS of the Venom

Eight micrograms of soluble venom was applied in an LC-MS system composed of an HPLC UltiMate 3000; Dionex, RSLCnano System (Thermo Fisher Scientific, San Jose, CA, USA) coupled to an LTQ-Orbitrap Velos mass spectrometer (Thermo Fisher Scientific, San Jose, CA, USA). Venom was fractionated through a 10-cm reversed-phase C18 in-house-made column (filled with Jupiter^®^ 4 µm Proteo 90 Å resin, Phenomenex, Torrance, CA, USA), using a linear gradient of 5% to 90% of solvent B (0.1% formic acid in acetonitrile) in 240 min, with a flow rate of 300 nL/min. The resolved peptides were ionized by a nano-electrospray ion source. A full scan MS was used (400–2000 m/z) with a resolution of 60,000. The monoisotopic molecular mass is reported for components below 3000 Da, and the average molecular masses for those above 3000 Da.

### 4.5. Identification of Venom Components by LC-MS/MS

Three hundred micrograms of the venom soluble fraction were reduced with dithiothreitol (DTT) 10 mM (Sigma–Aldrich, Saint Louis, MO USA) and alkylated with iodoacetamide (IAA) 55 mM (Sigma–Aldrich, Saint Louis, MO USA). After that, the samples were digested with trypsin (Promega Sequencing Grade Modified Trypsin; Madison, WI, USA) using a 1:50 enzyme:protein ratio by weight, in 50 mM ammonium bicarbonate buffer (ABC). The samples were acidified with 10 µL of 10% formic acid (FA). The tryptic peptides were desalted with Sep-Pak tC18 cartridges (Waters, Milford, MA, USA) following the manufacturer’s protocol, and dried in a SpeedVac (Savant SPD1010, Thermo Scientific, San Jose, CA, USA).

Samples containing 4 μg of tryptically digested venom in 10 μL of solution A (0.1% formic acid) were analyzed. The proteins were separated through a 15-cm in-house-made column (filled with the same C18, Jupiter 4 µm Proteo 90 Å resin) using a linear gradient of 5% to 75% of solvent B (0.1% formic acid in acetronitrile) in 270 min. Mass spectra were registered in a full scan of 350 m/z to 1400 m/z with a resolution of 60,000. The MS/MS spectra were analyzed with the Proteome Discoverer 1.4.1.14 suite (Thermo Fisher Scientific, San Jose, CA, USA), employing the Sequest HT search engine with the following parameters: two missed cleavages, dynamic modifications (methionine oxidation, glutamine and asparagine deamidation), static modifications (cysteine carbamidomethylation), precursor mass tolerance of 20 ppm, fragment mass tolerance 0.6 Da, and 1% false discovery rate. The database of the translated transcripts from Transdecoder was used for protein identification. An identification was considered positive when a minimum of two matching peptides were identified, and Sequest HT gave a global score (sum of all peptides’ XCorr) of at least 20.

## Figures and Tables

**Figure 1 toxins-11-00247-f001:**
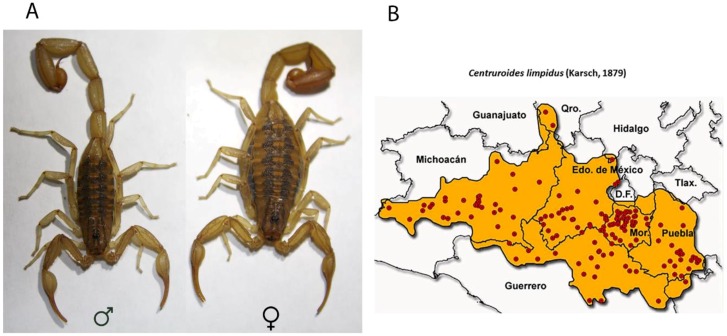
Habitus and distribution of *Centruroides limpidus*. (**A**) The morphology of *C. limpidus*, male (left) and female (right). (**B**) Geographical distribution of *C. limpidus* in 10 Mexican States (red dots indicate the places of sampling). (**B**) Reproduced with permission from [14] Copyright 2009, Universidad Nacional Autónoma de México.

**Figure 2 toxins-11-00247-f002:**
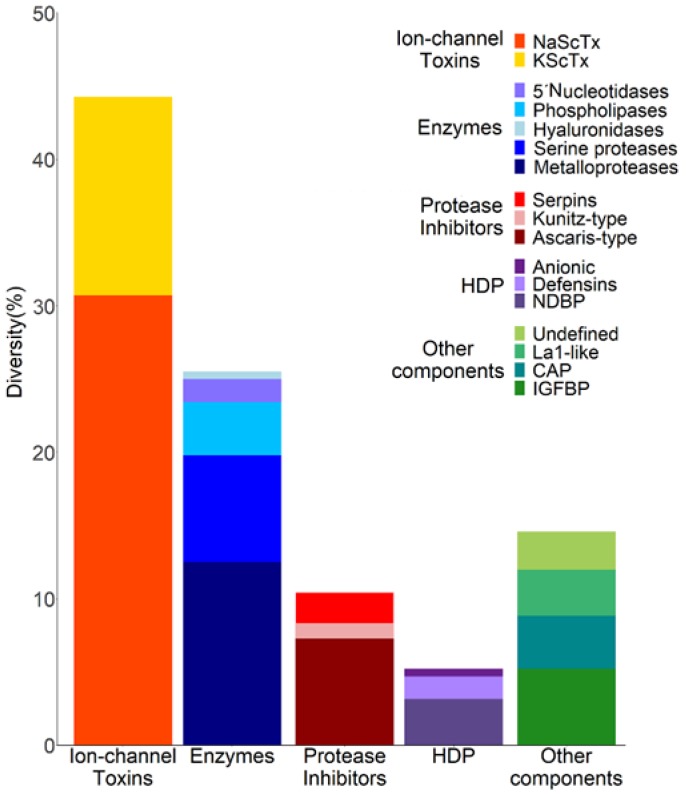
Relative diversity of transcripts related to venom components. For the graphic, only the number of different transcripts identified for each category.

**Figure 3 toxins-11-00247-f003:**
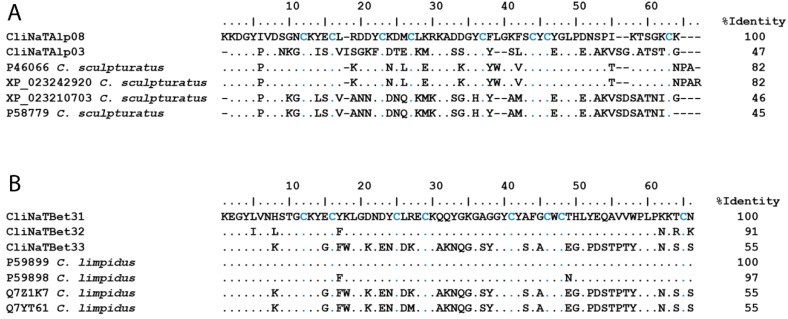
Examples of Na^+^ channel-specific toxins. (**A**) Putative α-NaScTx and (**B**) Putative β-NaScTx. The transcript-derived peptides are aligned to their reference sequences. The percentage of identity was calculated considering only the mature sequences. Accession numbers and species’ names of the references were taken from UniProt or GenBank. Conserved cysteine residues are highlighted in blue. Dots indicate identical residues, and dashes indicate gaps.

**Figure 4 toxins-11-00247-f004:**
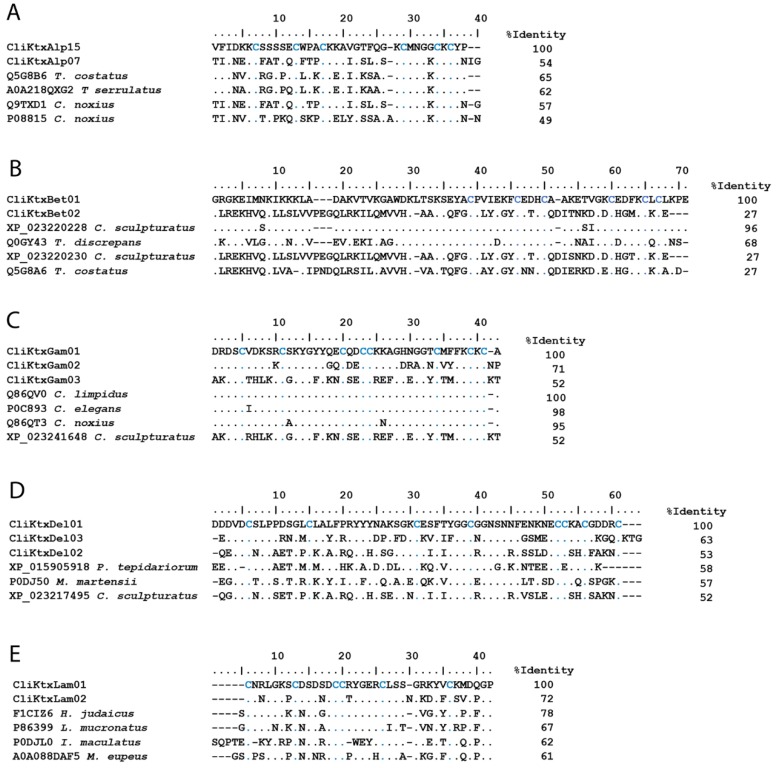
Examples of K^+^ channel-specific toxins. (**A**–**E**) show the alignments of the transcript-derived sequences of members of the α, β, γ, δ, and λ-KScTx families, respectively, with the reference sequences. The percentage of identity was calculated considering only the mature sequences. Accession numbers and species’ names of the references were taken from UniProt or GenBank. Conserved cysteine residues are highlighted in blue. Dots indicate identical residues, and dashes indicate gaps.

**Figure 5 toxins-11-00247-f005:**
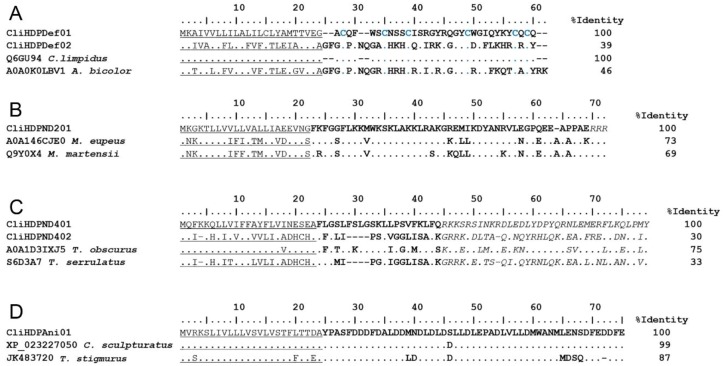
Examples of host defense peptides (HDPs). The translated sequences from representative transcripts coding for HDPs found in the *C. limpidus* transcriptome are shown aligned to matching sequences from databases. The complete precursor sequences are shown and the complete precursor was considered in the calculation of the percentage of identity. (**A**) Defensins. (**B**) The unique NDBP-2 precursor found. (**C**) Two putative NDBP-4 precursors. (**D**) The unique precursor for the putative anionic peptide. Accession numbers and species’ names of the references were taken from UniProt or GenBank. Conserved cysteine residues are highlighted in blue. The predicted mature sequences are indicated in bold typeface, the predicted signal peptides are underlined and the propeptides are indicated in italics. Dots indicate identical residues and dashes indicate gaps.

**Figure 6 toxins-11-00247-f006:**
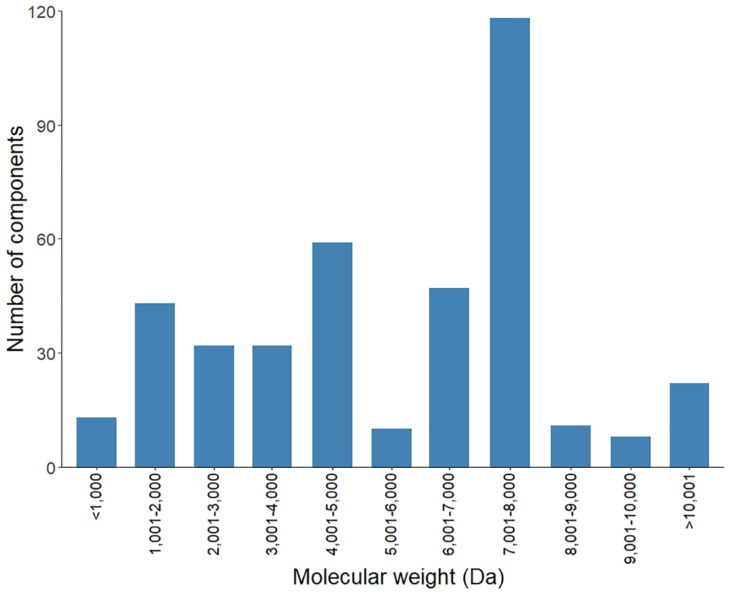
Distribution of the *C. limpidus* venom components detected by mass fingerprint with respect to their molecular masses.

**Figure 7 toxins-11-00247-f007:**
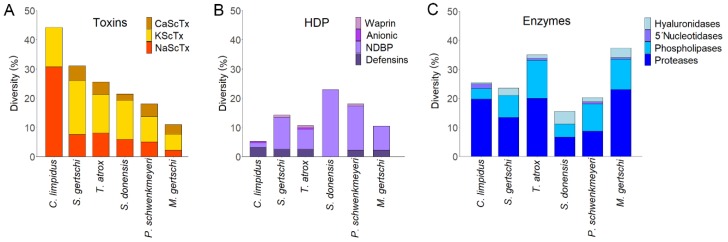
Comparative diversity of the transcripts coding for toxins (**A**), HDPs (**B**), and enzymes (**C**) found in *C. limpidus* versus those from non-buthid species reported in other transcriptomic analyses.

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
