# Peer review of "Dissecting Toxicity: The Venom Gland Transcriptome and the Venom Proteome of the Highly Venomous Scorpion Centruroides limpidus (Karsch, 1879)"

_toxins, 2019, doi:10.3390/toxins11050247_

Round 1

Reviewer 1 Report

The manuscript entitled “ Dissecting toxicity: the venom gland transcriptome 2 and the venom proteome of the highly venomous scorpion Centruroides limpidus (Karsch, 1879) deals with the  proteomic and transcriptomic characterization of the composition of the venom and venom gland mRNA of an important scorpion specie from Mexico. This study reported the identification of 192 venom toxins, including the toxins targeting Na+ and K+ channel-acting toxins, some enzymes, host defense peptides, and other venom components.

The experiments to get the results presented in the present manuscript were well conducted by the authors, nicely  presented and discussed by the authors. The discussion was relatively long, which makes the manuscript too much descriptive, despite to have been elegantly written. The manuscript as whole  identified a series of known and novel toxins in C. limpidus venom, which certainly will  bring a nice contribution to the toxinology of scorpions.

The manuscript only requires a minor intervention by the authors, in the introduction, which presents the Figure 1, that was not mentioned in the text.

Author Response

Reviewer 1:

Your commentary: “The manuscript entitled “ Dissecting toxicity: the venom gland transcriptome 2 and the venom proteome of the highly venomous scorpion Centruroides limpidus (Karsch, 1879) deals with the  proteomic and transcriptomic characterization of the composition of the venom and venom gland mRNA of an important scorpion specie from Mexico. This study reported the identification of 192 venom toxins, including the toxins targeting Na+ and K+ channel-acting toxins, some enzymes, host defense peptides, and other venom components.

The experiments to get the results presented in the present manuscript were well conducted by the authors, nicely  presented and discussed by the authors. The discussion was relatively long, which makes the manuscript too much descriptive, despite to have been elegantly written. The manuscript as whole  identified a series of known and novel toxins in C. limpidus venom, which certainly will  bring a nice contribution to the toxinology of scorpions.”

Answer: Thank you very much for your very nice comments on our manuscript. We do appreciate the time you have taken to review it. In the Results and Discussion section we tried to put emphasis on the components responsible for toxicity in this very dangerous scorpion species, highlighting the abundance of ion channel-acting toxins, but also enzymes, since they are not only cofactors for toxicity (acting as spreading factors), but can also be toxic by their own (as seems to be the case for antareases). The discussion became a bit long, we agree, but that’s because we wanted to cover all the toxic components that could contribute to this venom’s high toxicity, describing them by families and subfamilies to keep a comprehensive structure. We privileged giving all the information over writing a more succinct manuscript. But, except for the antimicrobial peptides (which are also discussed in detail due to the interest they attract for they potential applicability, and also to underscore their lower abundance with respect to non-buthid venoms), the rest of the many venom components are barely discussed, in particular those not representing any novelty. We grouped them all as “Other components” and most information on them is given in supplementary tables.

Your commentary: “The manuscript only requires a minor intervention by the authors, in the introduction, which presents the Figure 1, that was not mentioned in the text.”

Answer: We think the reviewer inadvertently missed the mention of Figure 1A and B in the introductory section. Please verify that they were indeed mentioned in lines 42 and 44 of the manuscript. Thank you!

Reviewer 2 Report

The authors present a remarkably comprehensive proteomic and transcriptomic analysis of the venom and venom gland of the particularly toxic scorpion species, Centruroides limpidus.  This species and 41 other species of its genus are responsible to the majority of envenomations in central Mexico as the organisms typically coexist within human environments.  A greater understanding of its armamentarium is pivotal to designing countermeasures against its sting.

After gathering several specimens from diverse areas and harvesting venom and venom glands via typical methods, the author subjected the materials to proteomic analyses with LCMS and assessed RNA harvested from glands.

The data generated demonstrated this species to be quite diverse in the category of venom compound, with numerous sodium and potassium channel blocking compounds identified, as well as other categories of compound not previously identified (e.g., 5’Nucleotidases).  Considered in composite, the ability of the venom to inflict paralysis in mammalian and other species is not surprising.

The authors also meticulously related their findings to the venom derived from other genus of scorpion, which is of evolutionary and potentially medicinal interest.

With regard to the serine proteases and metalloproteinases identified, would the authors please elaborate on their potential targets beyond pancreatitis or simple tissue injury?  Could they potentially affect coagulation based on the sequences identified?

In sum, the present work is of significant importance to those interested in obtaining a deeper understanding of the potential compounds at play during envenomation by this species.

Author Response

Reviewer 2:

Your commentary: “The authors present a remarkably comprehensive proteomic and transcriptomic analysis of the venom and venom gland of the particularly toxic scorpion species, Centruroides limpidus.  This species and 41 other species of its genus are responsible to the majority of envenomations in central Mexico as the organisms typically coexist within human environments.  A greater understanding of its armamentarium is pivotal to designing countermeasures against its sting.

After gathering several specimens from diverse areas and harvesting venom and venom glands via typical methods, the author subjected the materials to proteomic analyses with LCMS and assessed RNA harvested from glands.

The data generated demonstrated this species to be quite diverse in the category of venom compound, with numerous sodium and potassium channel blocking compounds identified, as well as other categories of compound not previously identified (e.g., 5’Nucleotidases).  Considered in composite, the ability of the venom to inflict paralysis in mammalian and other species is not surprising.

The authors also meticulously related their findings to the venom derived from other genus of scorpion, which is of evolutionary and potentially medicinal interest.”

Answer: Thank you very much for taking the time to make such a thorough and encouraging review. We are very much obliged!

Your commentary: “With regard to the serine proteases and metalloproteinases identified, would the authors please elaborate on their potential targets beyond pancreatitis or simple tissue injury?  Could they potentially affect coagulation based on the sequences identified?

Answer: This is a very interesting question that would need a lot of further discussion. There are many mechanisms which could affect coagulation and therefore hemostasis, and proteases surely can have a direct effect by themselves (e.g. by degrading components of the clotting pathways, such as the matrix proteins or fibrinogen). But they could also affect blood clotting in an indirect way. For example, the demonstrated secretagogue effect of antareases over pancreatic enzymes (e.g. amylase) could amplify this effect (amylase itself affects coagulation). One has to keep in mind that protease inhibitors are also present in scorpion venoms, and they can exert an inhibitory effect over procoagulant components (e.g. thrombin). So, the combined effect of proteases and inhibitors could indeed have an impact on coagulation. The problem is that, as far as we know, hemostasis issues are not really common in intoxications by this species, not even making the top list in the clinical profile for scorpion envenomation in Mexico (we reviewed this in Chavez-Haro, A.L. & Ortiz, E. 2014. Scorpionism and Dangerous Species of Mexico in: Rodriguez-de-la-Vega, R. Scorpion Venoms. Springer Netherlands. pags. 201-213). We are aware of a few examples of scorpion stings resulting in these issues in a small percentage of patients elsewhere (e.g. Sarkar, S. et al. Indian J Crit Care Med. 2008; 12(1): 15–17), with some events leading even to death (Dube, S at al. J Indian Med Assoc. 2011; 109(3): 194-5) and we do understand the relevance of your question. But, trying to assign activities to putative enzymes discovered from their coding mRNA sequences seems a bit speculative. We can only assume what function these enzymes could have by association with known effects of analogous, purified and characterized scorpion venom enzymes. If we were talking about snakes, then yes, we could probably imply the involvement of the metalloproteases in coagulation, but with the small amounts of venom injected in a scorpion sting, it would be too far-fetched, in our opinion. We hope the reviewer understands our point of view.

Your commentary: “In sum, the present work is of significant importance to those interested in obtaining a deeper understanding of the potential compounds at play during envenomation by this species.”

Answer: Again… Thank you very much!

Reviewer 3 Report

In this manuscript, the authors report a transcriptomic and proteomic analysis of the venom content of the scorpion Centruroides limpidus (Karsch, 1879). The analysis of 80 million reads led to identification of 192 putative venom-associated transcripts, mostly Na/K channel toxins, host defence proteins (HDP), proteases and other components. Proteomic analysis performed by LC-MS/MS could confirm 46 peptides identified by transcriptomic analysis.

 Overall, this manuscript is perfectly intelligible, the methodology looks appropriate, and the reported data (including the supplementary material) are very exhaustive.

 Minor comments:

The authors could assemble Illumina data into 198,662 putative transcripts, of which 11,058 were annotated by similarity to sequences from available databases. There is a relatively large portion of novel, unannotated sequence. It would be interesting if the authors could add some comments on this.

 Although the manuscript is perfectly intelligible, there are few odd sentences.

 Page 2, line 45: “It produces a potent venom which is highly toxic for mammals, and poses a serious threat to human life, having an LD50 of ca 15 μg/20g mouse”. Although it is clear that LD50 refers to the mouse, this sentence is not correct.

Page 3, line 77: “This section may be divided by subheadings. It should provide a concise and precise description 77 of the experimental results, their interpretation as well as the experimental conclusions that can be 78 drawn.” This sentence is not necessary

To conclude, I believe this manuscript fully meets the interests of the broad readership of Toxins. Therefore, I would recommend it for publication on Toxins.

Author Response

Reviewer 3:

Your commentary: “In this manuscript, the authors report a transcriptomic and proteomic analysis of the venom content of the scorpion Centruroides limpidus (Karsch, 1879). The analysis of 80 million reads led to identification of 192 putative venom-associated transcripts, mostly Na/K channel toxins, host defence proteins (HDP), proteases and other components. Proteomic analysis performed by LC-MS/MS could confirm 46 peptides identified by transcriptomic analysis.

 Overall, this manuscript is perfectly intelligible, the methodology looks appropriate, and the reported data (including the supplementary material) are very exhaustive.”

Answer: Thank you very much for your kind comments on our manuscript and for your valuable corrections. We have taken into account all your observations as indicated below.

Your commentary: “Minor comments:

The authors could assemble Illumina data into 198,662 putative transcripts, of which 11,058 were annotated by similarity to sequences from available databases. There is a relatively large portion of novel, unannotated sequence. It would be interesting if the authors could add some comments on this.”

Answer: It is indeed a problem, which is related to the absence of matching sequences in the databases (annotation is performed by sequence similarity) and demonstrates that the biochemical and functional characterization of the scorpion venoms is far from complete. We have added the following comment in the discussion as you suggest: “The divergence between the large number of assembled transcripts and the smaller subset of sequences with annotation, reflects the lack of information on many scorpion venom components, and reinforces the need for further biochemical and functional characterization of the scorpion venoms”. We hope you find it adequate.

Your commentary: “Although the manuscript is perfectly intelligible, there are few odd sentences.

 Page 2, line 45: “It produces a potent venom which is highly toxic for mammals, and poses a serious threat to human life, having an LD50 of ca 15 μg/20g mouse”. Although it is clear that LD50 refers to the mouse, this sentence is not correct.”

Answer: The reviewer is right. We have modified that sentence for the sake of clarity. It now reads: “It produces a potent venom which is highly toxic for mammals (the LD50 in mice is ca. 15 μg/20g), and poses a serious threat to human life”.

Your commentary: “Page 3, line 77: “This section may be divided by subheadings. It should provide a concise and precise description 77 of the experimental results, their interpretation as well as the experimental conclusions that can be 78 drawn.” This sentence is not necessary”

Answer: We are very sorry for this. That sentence was carried unmodified from the original template from Toxins, which is required for manuscript submission. It was now removed from the manuscript. Thank you.

Your commentary: “To conclude, I believe this manuscript fully meets the interests of the broad readership of Toxins. Therefore, I would recommend it for publication on Toxins.”

Answer: We do appreciate your favorable comments. Thank you very much!
